# Brain Metabolic Alterations in Seropositive Autoimmune Encephalitis: An ^18^F-FDG PET Study

**DOI:** 10.3390/biomedicines11020506

**Published:** 2023-02-09

**Authors:** Sébastien Bergeret, Cristina Birzu, Pierre Meneret, Alain Giron, Sophie Demeret, Clemence Marois, Louis Cousyn, Laura Rozenblum, Alice Laurenge, Agusti Alentorn, Vincent Navarro, Dimitri Psimaras, Aurélie Kas

**Affiliations:** 1Sorbonne University, AP-HP, Pitié Salpêtrière-Charles Foix Hospital Group, Nuclear Medicine Department, 75013 Paris, France; 2Paris Brain Institute, ICM, Sorbonne University, AP-HP, UMR S 1127, INSERM, Pitié Salpêtrière-Charles Foix Hospital Group, Service de Neurologie 2-Mazarin, 75013 Paris, France; 3Nuclear Medicine Department, Eugène Marquis Centre, INSERM, LTSI-UMR 1099, 35000 Rennes, France; 4Laboratoire d’Imagerie Biomédicale, LIB, Sorbonne Université, CNRS, INSERM, 75006 Paris, France; 5Sorbonne University, AP-HP, Pitié Salpêtrière-Charles Foix Hospital Group, Neurology Department, Neurological Intensive Care Unit, 75013 Paris, France; 6Sorbonne University, AP-HP, Pitié-Salpêtrière-Charles Foix Hospital Group, Epilepsy Unit, Paris Brain Institute, ICM, Reference Center for Rare Epilepsies, 75013 Paris, France; 7Sorbonne University, Laboratoire d’Imagerie Biomédicale, LIB, CNRS, INSERM, AP-HP, Pitié Salpêtrière-Charles Foix Hospital Group, Nuclear Medicine Department, 75013 Paris, France

**Keywords:** positron emission tomography, autoimmune encephalitis, diagnostic imaging, imaging biomarkers, metabolism

## Abstract

Introduction: Autoimmune encephalitis (AE) diagnosis and follow-up remain challenging. Brain ^18^F-fluoro-deoxy-glucose positron emission tomography (FDG PET) has shown promising results in AE. Our aim was to investigate FDG PET alterations in AE, according to antibody subtype. Methods: We retrospectively included patients with available FDG PET and seropositive AE diagnosed in our center between 2015 and 2020. Brain PET Z-score maps (relative to age matched controls) were analyzed, considering metabolic changes significant if |Z-score| ≥ 2. Results: Forty-six patients were included (49.4 yrs [18; 81]): 13 with GAD autoantibodies, 11 with anti-LGI1, 9 with NMDAR, 5 with CASPR2, and 8 with other antibodies. Brain PET was abnormal in 98% of patients versus 53% for MRI. The most frequent abnormalities were medial temporal lobe (MTL) and/or striatum hypermetabolism (52% and 43% respectively), cortical hypometabolism (78%), and cerebellum abnormalities (70%). LGI1 AE tended to have more frequent MTL hypermetabolism. NMDAR AE was prone to widespread cortical hypometabolism. Fewer abnormalities were observed in GAD AE. Striatum hypermetabolism was more frequent in patients treated for less than 1 month (*p* = 0.014), suggesting a relation to disease activity. Conclusion: FDG PET could serve as an imaging biomarker for early diagnosis and follow-up in AE.

## 1. Introduction

Autoimmune encephalitis (AE) consists of a rapidly progressive, immune mediated encephalopathy, without any evidence of infection. Two main physio pathological mechanisms are involved: (i) antibody mediated, when the targeted auto-antigen is located on the neural cell surface, and (ii) T-cell mediated, when the targeted neural auto-antigen is intracellular. The accurate diagnosis of AE remains a challenge in the clinical setting. The increasing therapeutic arsenal and an earlier treatment initiation have enabled an improvement in long-term neurological outcomes. However, the need for relevant biomarkers comprising biological (cytokines, antibodies) and imaging (MRI and FDG PET) parameters remains unsatisfied, despite expanding knowledge on this disease. Among imaging biomarkers, brain ^18^F-fluoro-deoxy-glucose PET (FDG PET) has shown promising results in AE with (i) higher sensitivity than that of brain MRI (87% vs. 56%) [1], and (ii) discrimination between AE patients and healthy controls with 98% specificity [2]. Moreover, cerebral metabolic patterns on FDG PET seem correlated with the detected autoantibody, with, for example: (i) more frequent hypermetabolism in mesial temporal regions and striatum [1,3,4,5] in leucine-rich glioma-inactivated 1 (LGI1) AE patients, and (ii) wedge-shaped occipital hypometabolism in N-methyl-D-aspartate receptor (NMDAR) AE [6,7,8,9]. Yet, there are contradictory results in the literature, and more research is needed to explore the association between brain metabolic patterns on FDG PET and clinical phenotypes, as well as the impact of immunomodulatory treatments.

The latest 2016 consensus diagnostic criteria for AE [10] enable a probabilistic diagnosis (i.e., before antibody testing results are available) of “possible auto-immune encephalitis” allowing anticipated treatment initiation, based on clinical, biological, EEG and MRI findings. However, FDG PET was not integrated in the criteria for “possible auto-immune encephalitis” and was included only as an alternative to MRI in the criteria for “definite limbic encephalitis”.

Since then, a growing body of evidence has been published concerning brain FDG PET, supporting its diagnostic value in AE. Moreover, FDG PET has become increasingly available. As a result, AE is now a common indication for neurological FDG PET studies [11]. 

Our aim was to expand the knowledge concerning the FDG PET brain metabolic pattern of AE patients. To this end, we investigated FDG PET alterations in a cohort of seropositive AE patients. We also investigated FDG PET alterations according to auto-antibody type, and according to the presence or absence of immunomodulating treatments received prior to FDG PET.

## 2. Materials and Methods

### 2.1. Patient Selection

We conducted a monocentric retrospective study of patients with seropositive AE, evaluated with FDG PET, from January 2015 to October 2021, at the Pitié-Salpêtrière Hospital, Paris, France. Patients were selected from the database of the regional reference center for AE at the Pitié-Salpêtrière hospital, Paris, France. The main inclusion criterion was a diagnosis of seropositive AE based on Graus criteria [10] as follows: (i) symptoms evocative of AE, with sub-acute onset, (ii) serum or CSF auto-antibody positivity, (iii) compatible findings on CSF testing and/or EEG and/or MRI and (iv) reasonable exclusion of alternative diagnoses.

Additional inclusion criteria were: (i) available FDG PET including brain images, and (ii) age over 18 years. Previous history of medical conditions which may impact brain metabolism on FDG PET (vascular, tumoral or neurodegenerative) constituted a non-inclusion criterion.

The use of our institutional FDG PET database for scientific research was ethically approved by the French authority for the protection of privacy and personal data in clinical research (CNIL, approval No. 2111722). As per French regulations, all subjects in this database were informed of the possible use of their clinical data for research purposes, and were given the opportunity to refuse that their data be used in this way. This study was performed in accordance with the principles of the Declaration of Helsinki.

### 2.2. Clinical, Biological and MRI Data

We recorded patient demographics, previous clinical history, the clinical phenotype at disease onset and at the time of the FDG PET, and the received immunomodulatory treatments. For each patient, we recorded whether or not a concurrent cancer diagnosis had been established. CSF parameters (cell count, intrathecal immunoglobulin synthesis) and brain MRI reports were collected when available. In the case of multiple MRI exams, we selected the one nearest to symptom onset.

### 2.3. FDG PET Imaging

FDG PET was performed in a standardized resting state (reduced ambient noise) with normal blood glucose levels and after at least 4 h of fasting. PET images were acquired 30–60 min and 60–90 min after injection of 2–5 MBq/kg of FDG, for brain and whole-body studies (for detection of malignancies), respectively. Either a dedicated brain acquisition or the combination of brain and whole-body acquisitions was performed according to the clinical context. From 2015 to 2021, 4 PET devices were used in our center: Gemini GXL PET/CT (Philips, Eindhoven, The Netherlands), Biograph mCT flow PET/CT (Siemens Healthineers, Erlangen, Germany), Signa 3T PET/MR (General Electrics Healthcare, Chicago, IL, USA) and Discovery MI5 PET/CT (General Electrics Healthcare, Chicago, IL, USA). Images were reconstructed using OSEM type iterative algorithms and corrected for photon attenuation with a CT scan for PET/CT and either ZTE sequence-based or combined Dixon and atlas-based methods for brain PET/MR. All the scans were reviewed to confirm completeness and to verify the quality of the acquisitions. When several FDG PET studies were available, we selected the one closest to symptom onset. 

Brain PET images were analyzed using Scenium suite software (Syngo.via version 5.1, Siemens Healthineers, Erlangen, Germany) which depicts cerebral FDG uptake and its statistical deviation from a healthy control database (Z-score map) [12,13,14]. To this end, datasets from Gemini GXL PET/CT were compared to a young or older adult normal FDG PET database (according to the age of the patient to be tested), acquired on the same PET device. For the other PET devices, we used 2 other FDG PET databases of healthy subjects acquired on the Biograph mCT flow PET/CT. All scans were rated for significant regional hypometabolism (i.e., Z-score value < −2) or hypermetabolism (i.e., Z-score value > 2) on individual Z-score maps, in standard brain regions. These regions included: (i) the prefrontal cortex (medial, dorsolateral, orbitofrontal areas), (ii) associative posterior cortex, (iii) primary cortices (visual, auditory, sensorimotor), (iv) medial temporal lobes (MTL), (v) cerebellum (hemispheres, dentate nuclei and vermis), (vi) brainstem (midbrain, pons, medulla oblonga), (vii) striatum and thalamus, in each hemisphere separately. 

As brain metabolism may vary with the stage of the disease and in response to treatment, we compared FDG PET results of the patients who were treatment-naïve at the time of PET, to those of the patients having already received treatment prior to PET. For a small subset of patients with available follow-up PET, we compared the PET profile before and after treatment. We also compared the demographic, clinical and imaging features according to autoantibody subtype. In patients for whom whole body PET images were acquired concomitantly to cerebral FDG PET, we recorded instances in which whole body PET revealed possible malignancy, based on the imaging reports. 

### 2.4. Statistical Analysis

Statistical analyses were performed using JMP software (JMP v16, SAS Institute Inc., Cary, NC, USA). Descriptive statistics included frequency as percentage for categorical variables and mean for continuous variables. Categorical variables were compared using chi-2 test, or the exact Fisher test. ANOVA or Welch’s ANOVA (considering unequal variances) was used for comparison of continuous variables as appropriate. A *p*-value < 0.05 was considered statistically significant (2-sided tests). Continuous variables are presented either as mean ± SD or mean [range].

## 3. Results

### 3.1. Patient Population

We identified 46 seropositive AE patients fulfilling the inclusion criteria (mean age at diagnosis 49.4 yrs [18; 81]; 59% women) with 44/46 definite limbic encephalitis (LE) according to the 2016 criteria [10]. Demographic characteristics, clinical, biological and MRI features are detailed in Table 1. Interestingly, women had a lower mean age at disease onset (42.1 yrs vs. 55.2 yrs, respectively, *p* = 0.004). 

Autoantibodies against neural surface antigens (NSA-ab) were present in 27 patients, including 11 with LGI1, 5 with Contactin-associated protein-like 2 (CASPR2), 2 with gamma-amino butyric acid receptor B (GABAR B), and 9 with NMDAR autoantibodies. Autoantibodies against intracellular antigens (ICA-ab) were detected in 18 patients, including glutamic acid decarboxylase (GAD, *n* = 13), Hu (*n* = 3), Ma2 (*n* = 1) and Amphiphysin (*n* = 1). One patient was positive for glial fibrillary acidic protein (GFAP) autoantibody (results are detailed in Table 1).

LGI1 patients were equally distributed gender-wise, while most patients with NMDAR and GAD autoantibodies were female (78% and 77%, respectively) and those with CASPR2 were all male (*p* = 0.01). NMDAR and GAD patients were younger (32.5 ± 13 yrs and 40.7 ± 13 yrs, respectively) compared to CASPR2 and LGI1 patients (67.4 ± 11 yrs and 69.4 ± 10 yrs, respectively, *p* = 0.0001). 

Eight patients fulfilled the 2021 criteria for paraneoplastic syndromes [15]. Of these patients, 2 had a concurrent teratoma (NMDAR *n* = 2), 3 had small cell lung cancer (Hu *n* = 2, GABAR B *n* = 1), 1 had seminoma (Ma2), 1 had breast cancer (Amphiphysin) and 1 had pancreatic cancer (GAD) (Table 1).

### 3.2. Symptoms at Disease Onset

The most frequent symptoms at disease onset were seizures, memory impairment, psychiatric symptoms and bradykinesia/bradypsychia. The prevalence of all other symptoms was less than 30% (Table 1). The prevalence of acute or subacute memory impairment was highest for CASPR2 (5/5) and LGI1 (73%) AE (Table 1). Seizures were frequent in LGI1 (90%), NMDAR (89%) and CASPR2 (4/5) AE. Bradykinesia/bradypsychia and psychiatric symptoms were most frequent in NMDAR AE (78% and 89%, respectively). Stiff person syndrome (SPS) was only reported in GAD AE (38%, *p* = 0.015). Cerebellar syndrome was most prevalent in GAD AE (*p* = 0.02). Differences between clinical profiles across autoantibody subtypes were not statistically significant (except for cerebellar syndrome and SPS). 

### 3.3. Brain MRI

MRI data was available for 45/46 patients. The mean time from disease onset to MRI was 19.1 months [0.1; 108]. MTL hyperintensities were reported on fluid-attenuated inversion recovery (FLAIR) MRI in 24/45 (53%) patients, of which 17 were bilateral. No other specific findings indicative of AE were observed on MRI. The proportion of MRI abnormalities was significantly higher for CASPR2 patients (5/5), followed by LGI1 and GAD and lowest for NMDAR patients (22%) (*p* = 0.05) (Table 1).

### 3.4. FDG PET Imaging

FDG PET results are detailed in Table 2. Concerning PET cameras, 17 studies were acquired on the Gemini GXL PET/CT, 18 on the Biograph mCT Flow PET/CT, 3 on the Discovery MI5 PET/CT, and 7 on the Signa 3T PET/MR.

Twenty-five patients were evaluated before initiation of immunomodulating treatment or after a short treatment duration (i.e., ≤1 month). For these patients, FDG PET was performed at an early stage of the disease (i.e., ≤4 months from symptom onset; mean 1.8 months [0.3; 4]) in 8 cases, 4 of whom had already received specific treatment. It was performed at a later phase (i.e., >4 months from symptom onset, mean 18.3 months [5; 83]) in 17 patients, of whom 6 patients had already received specific treatment.

The remaining 21 patients had received immunomodulatory treatments for more than 1 month prior to FDG PET. For 17 of these, the indication of FDG PET was treatment response assessment (mean time from treatment initiation of 18.2 months, [1.5; 72]). Other indications were (i) suspected relapse in 2 cases, or (ii) systematic follow-up of long lasting stable neurological symptoms in 2 cases. 

All patients but 1 still presented neurological symptoms at the time of FDG PET. For the latter (NMDAR AE), FDG PET was performed as systematic workup 2 months after immunomodulatory treatment discontinuation. The prevalence of neurological symptoms was lower at the time of FDG PET than at symptom onset, the most frequent being seizures (43%), memory impairment (43%), and psychiatric symptoms (33%) (Table 1).

Only 1 patient (GAD AE) had normal brain FDG PET, even though still symptomatic (temporal lobe focal seizures). Alterations in brain metabolism were seen in the remaining 45 patients (98%), most of them (76%) presenting with a co-occurrence of hypermetabolism in some regions and hypometabolism in others. Only few had exclusively hypermetabolism (7%) or hypometabolism (15%). The regions involved the most frequently were the MTL (76%), prefrontal cortex (74%), cerebellum (70%), posterior associative cortex (65%) and striatum (50%) (Table 2).

Increased glucose metabolism was observed in 43/46 patients (93%), which frequently involved the MTL (52%), striatum (43%) and cerebellum (41%) (Table 2). MTL hypermetabolism was bilateral in 14/24 patients (58%). Cerebellar hypermetabolism involved the dentate nuclei (30%) more frequently than cerebellar hemispheres (17%) or vermis (20%). Noteworthy, 12 patients (26%) displayed simultaneous hypermetabolism in the dentate nuclei and vermis. Cortical hypermetabolism was rare and concerned mainly the primary sensorimotor cortex (24%).

Cortical hypometabolism on the other hand was frequent (39/46 patients, 85%). It involved the prefrontal cortex (70%) and the posterior associative cortex (65%) in equal proportions. Widespread cortical hypometabolism (i.e., simultaneously involving the prefrontal and posterior associative cortices) was frequent (59%). FDG uptake was asymmetrical in 7 patients (15%). Hypometabolism was also observed in the cerebellum (43%), the MTL (24%), and the brainstem (22%).

Surprisingly, clinical symptoms did not correlate with the localization of metabolism alterations in limbic encephalitis; for instance, there was no significant association between MTL hypermetabolism and memory deficits (*p* = 0.4) or seizures (*p* = 0.6). However, patients with cerebellar syndromes tended to exhibit cerebellar hemisphere hypometabolism on PET more frequently (*p* = 0.07). The only patient without neurological symptoms showed mild hypometabolism in the prefrontal cortex, striatum, and cerebellum.

We compared the occurrence of MTL abnormalities on FDG PET to MTL FLAIR hyperintensities on brain MRI, in the 25 patients for whom the delay between PET and MRI was less than 2 months (GAD *n* = 6; LGI1 *n* = 6; NMDAR *n* = 5; CASPR2 *n* = 3; GABA B *n* = 1; Amphiphysin *n* = 1; GFAP *n* = 1; Ma2 *n* = 1; Hu *n* = 1). Overall, FDG PET was more sensitive than MRI (abnormal for 19/25 vs. 13/25 patients, respectively). FDG PET and MRI were in agreement (i.e., both abnormal or both normal) in 14/25 patients (56%). Abnormal findings were found on both FDG PET and MRI in 11/25 patients (44%), mainly consisting in MTL hypermetabolism (8/11, 73%) associated with MTL FLAIR hyperintensity. MTL were normal on both modalities in 3/25 patients (12%). FDG PET and brain MRI were in disagreement in the remaining 11/25 patients (44%): (i) 9/11 had abnormal PET without MR abnormalities (4 with MTL hypermetabolism, including 2 LGI1 and 2 NMDAR patients), and (ii) 3/11 patients had MTL FLAIR hyperintensity without FDG PET abnormalities (2 chronic GAD patients at 48 and 72 months of follow-up, with mild MRI abnormalities, and 1 LGI1 patient at 4 months of follow-up with PET abnormalities in other regions than the MTL).

### 3.5. FDG PET in LGI1, NMDAR, GAD and CASPR2 Antibody Subtypes

The most prevalent autoantibodies in our cohort were anti GAD, LGI1, NMDAR, and CASPR2 (*n* = 13, 11, 9 and 5, respectively, Table 1). The mean time from disease onset to FDG PET was 8.9 months [1; 24] for LGI1 and was longer for NMDAR (36.7 months [1; 108]), GAD (34.5 months [5; 84]) and CASPR2 (34.2 months [4; 83]) (*p* = 0.024). The mean time from treatment initiation to FDG PET was slightly longer for GAD and NMDAR (16 months [0; 72] and 31 months [0.1; 108], respectively) than CASPR2 and LGI1 (8 months [0; 39] and 2.5 months [0; 16], respectively) but without reaching statistical significance (*p* = 0.1). 

Brain metabolic profiles on FDG PET tended to differ between LGI1, NMDAR, GAD and CASPR2 AE without reaching statistical significance (Table 2, Figure 1). MTL abnormalities—mainly hypermetabolism—tended to be more frequent in CASPR2 (5/5) and LGI1 AE (9/11, 82%) than in GAD (7/13, 54%) and NMDAR AE (6/9, 67%). Cortical hypometabolism in NMDAR showed particular features, as: (i) it was more frequent (observed in all NMDAR patients), (ii) it involved a higher number of regions (*p* = 0.013), (iii) it was diffuse (i.e., affecting simultaneously the prefrontal and posterior associative cortices) in a higher proportion of patients (78%), (iv) it involved the occipital lobe more frequently (44% for NMDAR vs. 27% or less for others), and (v) severe “wedge-shaped” occipital hypometabolism was observed in the 2 NMDAR patients in the acute phase. Cerebellar abnormalities were more frequent in CASPR2 (5/5) and NMDAR (8/9, 89%) patients. Cerebellar hypometabolism had a similar prevalence among AE subtypes (approximately 40%). GAD patients had less frequent cerebellar hypermetabolism (15%) than others (up to 60%). Simultaneous hypermetabolism in the dentate nuclei and vermis was most frequent in CASPR2 AE (3/5). Striatum hypermetabolism had a similar prevalence among all AE subtypes (approximately 40%) but was more severe in LGI1 AE when present (mean Z-score 6.3 ± 1.5 in LGI1 AE versus up to 5.2 in others, *p* = 0.047). GAD patients had fewer abnormalities, with an overall lower number of pathological regions (*p* = 0.04).

### 3.6. Treated vs. Untreated Patients

Concerning immunomodulating drugs, 16 patients were treatment-naïve, while 30 patients had already received one or more treatments (IgIV 96%, steroids 73%, rituximab 60%, plasma exchange 23%, and cyclophosphamide 53%) prior to FDG PET. The untreated group presented a slightly higher frequency of: (i) hypometabolism of the prefrontal cortex (81% vs. 63% respectively), (ii) MTL hypermetabolism (69% vs. 43% respectively) and (iii) striatum hypermetabolism (56% vs. 37%), without reaching statistical significance. 

We also compared the FDG PET profile of a first group of patients having received none or less than 1 month of treatment (i.e., 16 untreated and 9 treated for less than 1 month) to a second group comprising the remaining 21 patients having received more than 1 month of treatment (Figure 2). Noteworthy, the first group comprised more LGI1 patients and fewer NMDAR patients (*p* = 0.02). The first group had a significantly higher number of abnormal regions (*p* = 0.02), and presented significantly more frequent striatal hypermetabolism (*p* = 0.014). They also tended to have (i) more frequent cortical hypometabolism (88% vs. 66% respectively), (ii) equally frequent but more severe MTL hypermetabolism (56% vs. 47% respectively; mean Z-score 7.0 vs. 4.7 respectively), (iii) slightly more frequent cerebellar hemisphere hypometabolism (48% vs. 33% respectively), and (iv) slightly more frequent dentate nuclei hypermetabolism (40% vs. 19%), without reaching statistical significance.

Five patients (LGI1, *n* = 3; GAD, *n* = 1; CASPR2, *n* = 1) had available follow-up FDG PET studies performed after immunomodulating treatment and acquired on the same PET device as the initial study (Figure 3). The initial PET studies showed combinations of striatum and MTL hypermetabolism, cortical hypometabolism and cerebellar abnormalities. The follow up studies, performed at 6 months, 7 months, 8 months (*n* = 2) and 5 years after the first PET showed clear improvement of PET abnormalities which were (i) almost normalized in 3 patients (2 LGI1 and 1 GAD), and (ii) completely normalized in 2 patients (1 LGI1 and CASPR2). FDG PET improvement was associated with (i) clinical improvement in modified Rankin Scale (mean mRS of 3.2 (3–4) initially versus 1.4 (1–2) at follow-up PET), and (ii) negative seroconversion in the 3 patients with LGI1 AE.

### 3.7. Whole Body FDG PET

Whole body PET studies were performed in 11/46 patients. Abnormal findings were reported in 3/11 patients (27%), of whom 2 patients had paraneoplastic forms, and for whom FDG PET showed pulmonary lesions with high FDG uptake, related to small cell lung cancer. The remaining patient had nasopharyngeal hypermetabolism and a lesion of the uterus, which were ultimately classified as benign abnormalities during follow-up. 

Of the 8 patients with normal PET, 2 patients had paraneoplastic forms (pancreatic adenocarcinoma and ovarian teratoma), but these patients had already received curative surgery before FDG PET. The remaining 6 patients did not present evidence of malignancy during follow-up.

## 4. Discussion

Our findings support the higher sensitivity of FDG PET compared to brain MRI in AE diagnosis in line with previous data [1,3,8,9,16,17]. Cerebral FDG PET allows the investigation of regional brain glucose metabolism alterations, translating several pathophysiological mechanisms including: (i) changes in neuronal and synaptic activity, (ii) astrocytic metabolism in relation to neuronal activity, (iii) glucose transport across the blood-brain barrier, and (iv) possibly microglial activation in inflammatory states [11]. This might explain the higher sensitivity of FDG PET when compared to brain MRI, which mainly depicts structural/morphological abnormalities. Indeed, in our cohort, almost all patients but 1 (98%) had abnormalities on FDG PET vs. 53% on MRI, using a semi-quantitative analysis of PET, a methodology which increases sensitivity [4,18].

In our cohort, the most frequently detected FDG PET pattern was the association of (i) regional brain hypermetabolism mostly involving the MTL and/or basal ganglia (65% of cases), and (ii) cortical hypometabolism (prefrontal and/or posterior associative cortices) (85% of cases). Cerebellar metabolism was also frequently abnormal (70% of cases). These glucose metabolism abnormalities are described in the literature across many AE types impervious to the detected antibody [1]. For instance, MTL and basal ganglia hypermetabolism has been reported in LGI1 (69% and 71%), GAD (21% and 10%) and NMDAR (11% and 62%), in association with diffuse cortical hypometabolism (9%, 35% and 70% for LGI1, GAD and NMDAR, respectively [1]). Interestingly, glucose cerebral hypermetabolism presented with higher Z-score amplitudes (mean Z-scores ranging from 4.8 to 6.5 in the MTL, cerebellar hemispheres, and striatum) than hypometabolism (mean Z-scores ranging from −3.5 to −4.6 in cortical regions, the MTL and the cerebellum). Moreover, the potential diagnostic value of the FDG PET cortex/striatum metabolic ratio has been highlighted recently in both seropositive and seronegative AE [2].

Our study reinforces the correlation between the FDG PET metabolic profile and a specific autoantibody in AE patients. This was expected from a physiopathological standpoint given the functional nature of FDG PET [2] and the NsAb pathogenic mechanisms which go beyond inflammation [19,20,21]. In our cohort, the anti LGI1/CASPR2 group displayed similar clinical profiles mainly consisting in seizures, memory impairment and behavioral changes. On FDG PET, they were more prone to MTL involvement (mostly hypermetabolism), in line with the literature (70% of LGI1 patients and almost all CASPR2 patients published) [1,3,4,5,18,22,23,24,25]. Cortical hypometabolism of the prefrontal cortex has also been described in LGI1 [5,26] and CASPR2 [18,22,25] patients, and was frequent in our cohort (64% in LGI1 and 3/5 in CASPR2 AE in our patients). This may relate to irritability, aggressiveness, or disinhibition, present in 50% of cases at disease onset in our cohort. Interestingly, hypermetabolism of the basal ganglia, another main feature in LGI1/CASPR2 AE, was not as frequent as expected in our patients (45%) [1,18,23,24] but was more severe in LGI1 AE than other subtypes (*p* = 0.047). Hypermetabolism of the motor cortex, another expected feature of LGI1 AE [5,22,26,27] was also not frequent in our cohort (only 18%). This could possibly be explained by treatment initiation prior to FDG PET. In CASPR2 AE, FDG PET abnormalities frequently involved the cerebellum, with concomitant vermis and dentate nuclei hypermetabolism (3/5), yet the correlation of this PET feature to the clinical profile remains unclear.

In NMDAR AE, FDG PET was abnormal in all patients, contrasting with the lowest MRI abnormality rate in our cohort, as already reported in this subtype [7,8,9,16,17]. This underscores the added-value of PET imaging in NMDAR AE. NMDAR autoantibodies down-modulate NMDAR signaling [28], and their pathogenicity goes beyond the modification of neuronal activity by local inflammation. NMDAR AE patients typically present with occipital hypometabolism in the acute phase [6,7,8,9], which may be related to high NMDAR expression in the occipital lobes [29]. The metabolic profile is then thought to evolve in a sequential manner, with (i) diffuse mild cortical hypometabolism in the partial recovery phase, and (ii) normalization of FDG PET with full clinical recovery [30]. Our results are consistent with this description, as the 2 patients in the acute phase presented severe “wedge-shaped” occipital hypometabolism while the 7 remaining patients had diffuse cortical hypometabolism. Interestingly, brainstem FDG PET abnormalities were present in 67% of these patients. 

In our cohort, patients with GAD AE presented fewer metabolic abnormalities than in other subtypes, in line with published data [1]. GAD AE stands out from a pathogenic standpoint, as it belongs to the ICA-ab family but is rarely associated with malignancy (conversely to other ICA-ab), and is associated with other systemic autoimmune diseases such as diabetes mellitus type 1. We observed several distinct clinical phenotypes in these patients including (i) status epilepticus, (ii) cerebellar ataxia, (iii) SPS, and (iv) limbic encephalitis, as already described [31]. Our GAD AE patients presented more frequent cortical hypometabolism (rarely described before [32,33,34]) less frequent MTL hypermetabolism (31% in our series versus 59% in previous publications [1]) and more frequent striatum hypermetabolism (38% in our series versus 10% in previous publications [1]) than expected. FDG PET was in line with the clinical phenotype in these patients: (i) 3/4 with cerebellar syndromes had cerebellar hypometabolism, and (ii) there was a higher prevalence of concurrent cortical hypometabolism with striatum and MTL hypermetabolism in epilepsy and psychiatric GAD AE patients. Interestingly, although some data has suggested that SPS may result from hyperexcitation of motor and premotor cortex [35], our 5 SPS GAD patients had metabolic changes similar to other GAD patients, and only 1/5 had motor cortex hypermetabolism.

Our study suggests striatal hypermetabolism as a disease activity marker. Basal ganglia hypermetabolism is a well-known feature of AE. In our patients, striatal hypermetabolism was less frequent in patients treated for more than 1 month (except for GAD AE, with similar striatum hypermetabolism in both groups) thus reinforcing previous data [3,26]. Trends for less frequent abnormalities in treated patients were also observed in almost all other territories. In our patients with available follow-up PET, we observed normalization of PET abnormalities including MTL hypermetabolism, cortical hypometabolism and cerebellar abnormalities, demonstrating post-treatment reversibility of abnormalities in regions other than the striatum, as previously reported [3,26,30,36]. Overall, our data show that FDG PET abnormalities are reversible after treatment, and that striatum hypermetabolism could constitute an early marker of treatment response. Further investigation is warranted, for validation through prospective longitudinal studies. 

The main limitation in our study was the retrospective design. This explains the inconsistencies in timing of FDG PET and brain MRI, with important delays from symptom onset in some cases, and lack of some clinical data (such as seizure-free interval before the FDG PET). As our patient recruitment originated from a regional reference center, many patients were evaluated in the context of secondary assessments, with longer delays from symptom onset. This particular recruitment may also explain the slightly higher rate of abnormal PET in our cohort, as patients consulting in our center may have suffered from more severe phenotypes. The long delays may have impacted (i) the metabolic pattern of some AE patients, and (ii) the sensitivity gap between PET and MRI. Another important limitation was the use of different PET cameras. We generated Z-scores with several different normal databases, and chose the appropriate one for each patient according to PET device and age range, which is expected to mitigate these potential biases. Another limitation was the small sample size in each antibody subgroup, which probably hindered the detection of statistically significant differences among subgroups. The lack of statistical power may also explain the surprising absence of statistical correlation between FDG PET findings and clinical symptoms at the time of FDG PET. In future, prospective longitudinal studies with larger patient samples are needed to mitigate these limitations, including control patients to assess PET specificity. 

## 5. Conclusions

Our study including 46 seropositive AE patients explored with FDG PET reinforces the previously reported higher sensitivity of FDG PET compared to MRI in AE patients. Moreover, we report differences in brain glucose metabolic profiles according to antibody subtype, expanding the knowledge from previously published antibody-specific PET findings. Additionally, we propose striatum hypermetabolism as a potential early imaging biomarker for treatment response assessment in NsAb AE. Therefore, FDG PET seems useful for the diagnosis and follow-up of AE and should be considered as a first-line imaging modality. Prospective and longitudinal data are, however, still needed, to establish and standardize the performances of FDG PET, and its added-value compared to current diagnostic criteria.

## Figures and Tables

**Figure 1 biomedicines-11-00506-f001:**
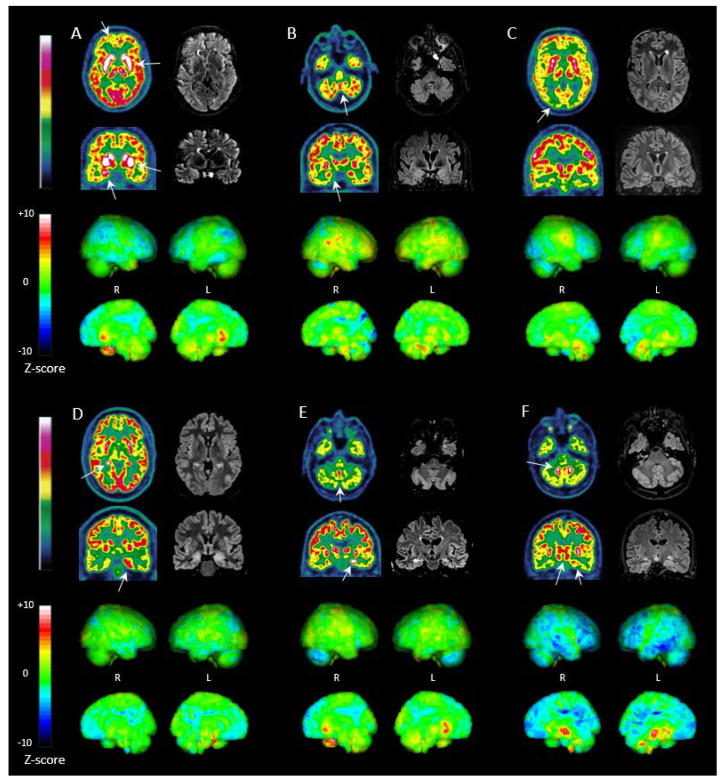
Examples of brain FDG PET and MRI FLAIR images in different AE subtypes. **Upper rows:** Axial and coronal sections of FDG PET (**left**) and FLAIR sequence of MRI (**right**) in patients with AE (white arrows indicate the main metabolic abnormalities). **Lower rows:** PET quantification (individual Z-score map relative to healthy age-matched controls) displayed as 3D statistical surface projections, lateral views (**top**) and medial views (**bottom**). (**A**) LGI1 AE with severe striatum hypermetabolism, bilateral MTL hypermetabolism, and cortical hypometabolism involving the prefrontal cortex on PET, with right MTL FLAIR hyperintensity and swelling on MRI. (**B**) CASPR2 AE with bilateral MTL hypermetabolism and increased vermis and dentate nuclei metabolism on PET, with bilateral MTL FLAIR hyperintensity and swelling on MRI. (**C**) NMDAR AE with severe occipital hypometabolism and moderate diffuse cortical hypometabolism on PET, and normal FLAIR images on MRI. (**D**) GAD AE with bilateral MTL hypermetabolism on PET (including the para-hippocampal gyrus and hippocampus), with bilateral MTL FLAIR hyperintensity and swelling on MRI. (**E**) Anti-Hu AE with bilateral MTL hypermetabolism and increased vermis and dentate nuclei metabolism, contrasting with cerebellar hemisphere hypometabolism on PET. MRI shows bilateral MTL FLAIR hyperintensity without swelling. (**F**) Anti-Ma2 AE with hypermetabolism in the striatum, MTL, brainstem, and dentate nuclei, contrasting with diffuse cortical hypometabolism and cerebellar hemisphere hypometabolism on PET. MRI shows bilateral MTL FLAIR hyperintensity with left MTL swelling. FDG PET: ^18^F-fluoro-deoxy-glucose positron emission tomography, MRI: Magnetic resonance imaging, FLAIR: Fluid attenuated inversion recovery, AE: autoimmune encephalitis, LGI1: Leucine-rich glioma-inactivated 1, MTL: medial temporal lobe, CASPR2: Contactin-associated protein-like 2, NMDAR: N-methyl-D-aspartate receptor, GAD: Glutamic acid decarboxylase.

**Figure 2 biomedicines-11-00506-f002:**
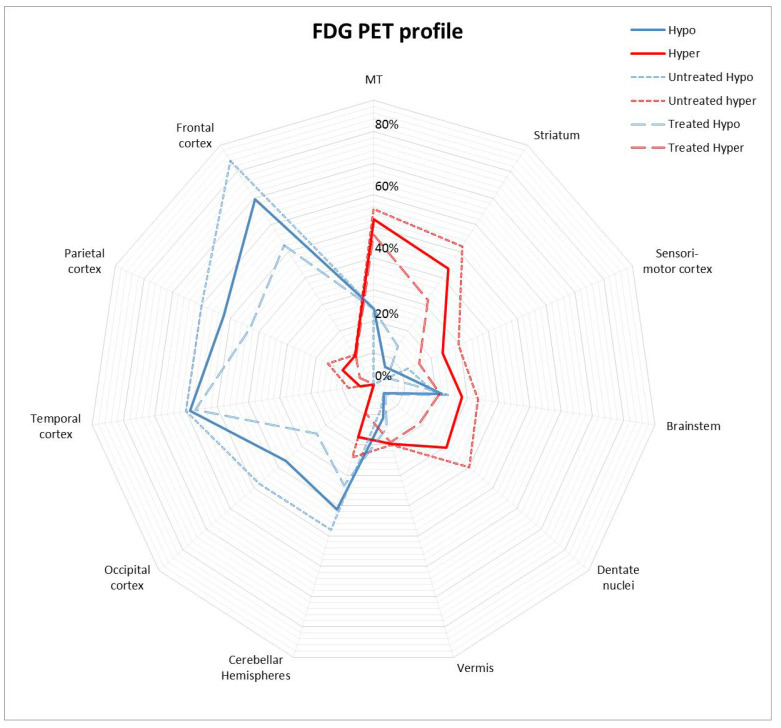
Prevalence of regional metabolic changes on FDG PET (hypermetabolism and hypometabolism) in the whole cohort, in patients treated for less than one month (referred to as untreated) and in those treated for more than one month (referred to as treated). FDG PET: ^18^F-fluoro-deoxy-glucose positron emission tomography, MT: medial temporal.

**Figure 3 biomedicines-11-00506-f003:**
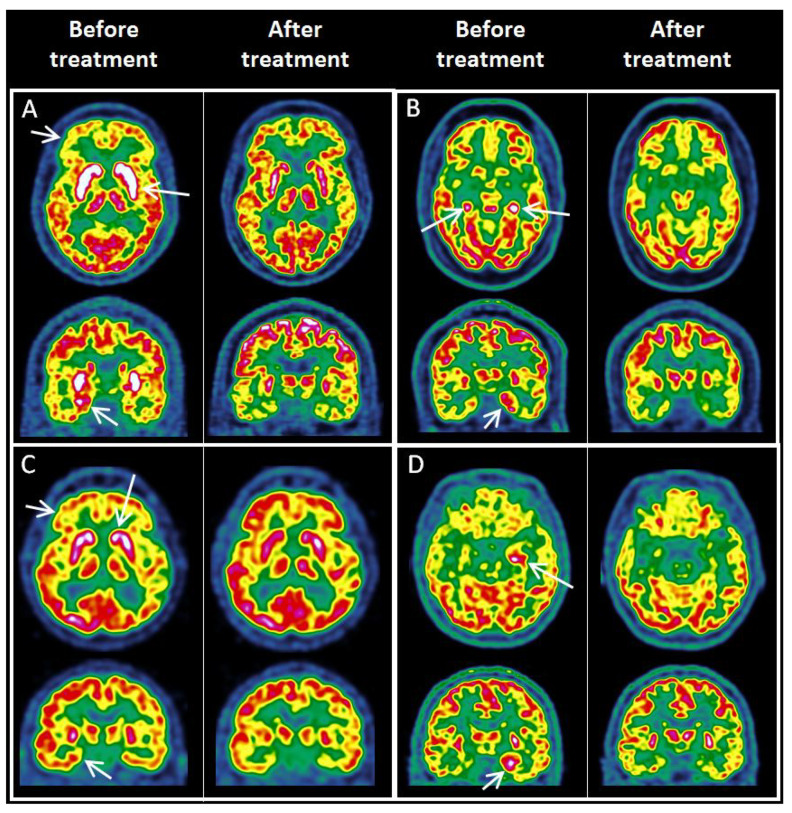
Examples of FDG PET images (axial and coronal sections) in patients with available post-treatment follow-up PET. White arrows indicate main metabolic abnormalities. (**A**) Patient with LGI1 AE, with initial PET showing striatum and MTL hypermetabolism and prefrontal cortex hypometabolism, almost normalized on follow-up PET. (**B**) Patient with anti-GAD AE, with initial PET showing bilateral MTL hypermetabolism, almost normalized on follow-up PET. (**C**) Patient with LGI1 AE, with initial PET showing striatum and MTL hypermetabolism and prefrontal cortex hypometabolism, completely normalized on follow-up PET. (**D**) Patient with LGI1 AE, with initial PET showing left MTL hypermetabolism, completely normalized on follow-up PET. FDG PET: ^18^F-fluoro-deoxy-glucose positron emission tomography, LGI1: Leucine-rich glioma-inactivated 1, GAD: Glutamic acid decarboxylase, MTL: medial temporal lobe.

**Table 1 biomedicines-11-00506-t001:** Demographic, clinical, CSF, and MRI finding across autoantibody subtypes.

	Total *n*= 46	GAD *n* = 13	LGI1 *n* = 11	NMDAR *n* = 9	CASPR2 *n* = 5	Hu *n* = 3	GABArB *n* = 2	Ma2 *n* = 1	Amphyphisin *n* = 1	GFAP *n* = 1	*p*-Value *
**Onset age (yrs), *mean [range]***	49.4 [19; 81]	38 [14; 59]	68.6 [43; 79]	29.8 [11; 55]	64.6 [54; 77]	15; 59;62	19; 73	29	47	41	
**M/F**	19/27	3/10	6/5	2/7	5/0	1/2	1/1	M	F	F	*0.01*
**Symptoms: *Inaugural n (%)*** ** *At time of PET n (%)* **											
**Memory impairment**	29 (63%)	6 (46%)	8 (73%)	5 (56%)	5/5	2/3	2/2	Yes	No	No	*NS*
	20 (43%)	6 (46%)	5 (45%)	1 (11%)	5/5	2/3	0	No	Yes	No	
**Seizures**	35 (76%)	8 (62%)	10 (91%)	8 (89%)	4/5	2/3	2/2	No	No	Yes	*NS*
	20 (43%)	6 (46%)	5 (45%)	3 (33%)	3/5	1/3	1/2	No	No	Yes	
**Psychiatric symptoms**	26 (57%)	5 (38%)	6 (55%)	8 (89%)	3/5	2/3	1/2	Yes	No	No	*NS*
	15 (33%)	4 (31%)	5 (45%)	3 (33%)	3/5	2/3	0	No	Yes	No	
**Bradypsychya/Bradykinesia**	26 (57%)	3 (23%)	3 (27%)	7 (78%)	2/5	2/3	1/2	Yes	Yes	No	*0.06*
	6 (13%)	0	0	1 (11%)	0	1/3	2/2	Yes	Yes	No	
**Cerebellar syndrome**	5 (11%)	4 (31%	0	0	0	1/3	0	No	Yes	No	*0.04* *0.02*
	5 (11%)	5 (38%)	0	0	0	0	0	No	Yes	No	
**Stiff limb syndrome**	5 (11%)	5 (38%)	0	0	0	0	0	No	No	No	*0.01*
	4 (9%)	4 (31%)	0	0	0	0	0	No	No	No	
**Abnormal movement**	5 (11%)	0	3 (27%)	1 (11%)	0	0	1/2	No	No	No	*NS*
	2 (4%)	0	0	2 (22%)	0	0	0	No	No	No	
**Speech impairment**	4 (9%)	0	0	2 (22%)	0	1/3	0	No	Yes	No	*0.09*
	2 (4%)	1 (8%)	0	1 (11%)	0	0	0	No	No	No	
**Sleep disorder**	14 (30%)	2 (15%)	5 (45%)	2 (22%)	3/5	1/3	0	Yes	No	No	*NS*
	1 (2%)	0	0	1 (11%)	0	0	0	No	No	No	
**Dysautonomia**	5 (11%)	1 (8%)	0	2 (22%)	1/5	0	0	Yes	No	No	*NS*
	2 (4%)	0	1 (9%)	0	0	0	0	No	Yes	No	
**Cancer *n (%)***	8 (17%)	1 pancreas cancer	0	2 Teratoma	0	2 Small cell lung cancer	1 Small cell lung cancer	Seminoma	Breast cancer	No	*NS*
**CSF *n (%)***											
**Auto-antibody location **** **CSF** **serum** **Both**	21 (47%) ** 16 (36% 8 (18%)	2 6 5	4 6 1	8 0 1	4 0 1	0 ** 2 0	2 0 0	0 1 0	0 1 0	1 0 0	*0.003*
**Pleiocytosis ****	16/26 (62%)**	4/8 (50%) **	2/6 (33%) **	4/5 **	1/2 **	2/2 **	1/1 **	NA **	Yes	Yes	*NS*
**Oligoclonal bands ****	7/16 (44%) **	4/8 (50%) **	1/5 **	2/3 **	0/1 **	NA **	1/1 **	NA **	NA **	Yes	*NS*
**MRI**											
**Time from symptom onset *mean [range], in months* ****	19.1 [0.1; 108] **	32.6 [0.1; 84]	5.7 [0.1; 24]**	22.4 [0.1; 108]	26.6 [0.1; 83]	2,4,26	0.6,4	3	5	0.1	*NS*
**MTL FLAIR hyperintensity** ***n (%) B,L,R* ****	24/45 (53%) 17,5,2 **	7 (54%) 7,0,0	6/10 (54%) 4,1,1 **	2 (22%) 2,0,0	5/5 3,2,0	2/3 1,0,1	1 (50%) 0,1,0	Yes L	No	No	*0.045*

* GAD versus LGI1 versus NMDAR versus CASPR2 (global comparison); ** missing values. CSF: cerebrospinal fluid, MRI: magnetic resonance imaging, GAD: Glutamic acid decarboxylase, LGI1: Leucine-rich glioma-inactivated 1, NMDAR: N-methyl-D-aspartate receptor, CASPR2: Contactin-associated protein-like 2, GABArB: Gamma-amino butyric acid receptor B, GFAP: Glial fibrillary acidic protein, FLAIR: Fluid attenuated inversion recovery.

**Table 2 biomedicines-11-00506-t002:** Brain FDG PET results in the whole cohort and across autoantibody subtypes.

	Total *n* = 46	GAD *n* = 13	LGI1 *n* = 11	NMDAR *n* = 9	CASPR2 *n* = 5	Hu *n* = 3	GABArB *n* = 2	Ma2 *n* = 1	Amphyphisin *n* = 1	GFAP *n* = 1	*p*-Value *
**Time from symptom onset** ** *mean [range], in months* **	24.4 [0.3; 108]	34.5 [5; 84]	8.9 [1; 24]	36.7 [1; 108]	34.2 [4; 83]	6; 12; 44	1; 1	5	6	0.3	*NS*
**Treated before PET *n (%)***	30 (65%)	7 (54%)	7 (64%)	8 (89%)	2/5	3/3	1/2	Yes	Yes	No	*NS*
**Treatment duration** ** *mean [range], in months* ** ******	12.3 [0; 108]	16.4 [0; 72]	2.5 [0; 16]	31 [0.1; 108] **	8.4 [0; 39]	1; 8; 13	0; 0.5	1	1	0	*NS*
**MTL, *n (%) B,L,R***											
**Hypermetabolism**	24 (52%) 14,2,8	4 (31%) 3,0,1	7 (64%) 2,4,1	3 (33%) 2,1,0	4/5 2,1,1	3/3 2,0,1	2/2 2,0,0	Yes B	No	No	*NS*
**Hypometabolism**	11 (24%) 5,5,1	3 (23%) 1,2,0	2 (18%) 0,1,1	3 (33%) 2,0,1	1/5 0,1,0	0	0	No	Yes B	Yes B	*NS*
**Striatum, *n (%)***											
**Hypermetabolism**	20 (43%)	5 (38%)	5 (45%)	4 (44%)	2/5	1/3	2/2	Yes	No	No	*NS*
**Hypometabolism**	3 (7%)	1 (8%)	0	1 (11%)	0	1/3	0	No	No	No	*NS*
**Cortical hypometabolism, *n (%)***											
**Frontal**	32 (70%)	7 (54%)	7 (64%)	8 (89%)	3/5	2/3	2/2	Yes	Yes	Yes	*NS*
**Parietal**	24 (52%)	5 (38%)	4 (36%)	7 (78%)	1	2/3	2/2	Yes	Yes	Yes	*NS*
**Temporal**	27 (59%)	6 (46%)	5 (45%)	7 (78%)	3/5	2/3	0	Yes	Yes	Yes	*NS*
**Occipital**	17 (37%)	1 (8%)	3 (27%)	4 (44%)	1/5	1/3	1/2	Yes	Yes	Yes	*NS*
**Asymetrical pattern**	7 (15%)	2 (15%)	0	2 (22%)	0	1/3	1/2	No	No	Yes	*NS*
**Anteroposterior gradient**	12 (26%)	1 (8%)	4 (36%)	2 (22%)	1/5	1/3	1/2	Yes	Yes	No	*NS*
**Sensorimotor hypermetabolism, *n (%)***	11 (23%)	2 (15%)	2 (18%)	3 (33%)	1/5	1/3	0	No	Yes	Yes	*NS*
**Cerebellar hemispheres, *n (%)***											
**Hypermetabolism**	8 (17%)	0	2 (18%)	2 (22%)	2/5	0	1/2	No	No	Yes	*NS*
**Hypometabolism**	19 (41%)	5 (38%)	4 (36%)	4 (44%)	2/5	2/3	0	Yes	Yes	No	*NS*
**Vermis, *n (%)***											
**Hypermetabolism**	9 (20%)	1 (8%)	3 (27%)	2 (22%)	2/5	0	1/2	No	No	No	*NS*
**Hypometabolism**	5 (11%)	2 (15%)	0	2 (22%)	1/5	0	0	No	No	No	*NS*
**Dentate nuclei, *n (%)***											
**Hypermetabolism**	14 (30%)	1 (8%)	3 (27%)	3 (33%)	3/5	2/3	1/2	Yes	No	No	*NS*
**Hypometabolism**	2 (4%)	0	0	1 (11%)	0	0	0	No	No	Yes	*NS*
**Brainstem, *n (%)***											
**Hypermetabolism**	13 (28%)	2 (15%)	3 (27%)	2 (22%)	2/5	2/3	1/2	Yes	No	No	*NS*
**Hypometabolism**	10 (22%)	3 (23%)	1 (9%)	4 (44%)	0	0	0	Yes	No	Yes	*NS*

* GAD versus LGI1 versus NMDAR versus CASPR2 (global comparison); ** missing values. FDG PET: ^18^F-fluoro-deoxy-glucose positron emission tomography, GAD: Glutamic acid decarboxylase, LGI1: Leucine-rich glioma-inactivated 1, NMDAR: N-methyl-D-aspartate receptor, CASPR2: Contactin-associated protein-like 2, GABArB: Gamma-amino butyric acid receptor B, GFAP: Glial fibrillary acidic protein, MTL: medial temporal lobe.

## Data Availability

Data is available upon reasonable request to the corresponding author.

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
