# Peer review of "Brain Metabolic Alterations in Seropositive Autoimmune Encephalitis: An 18F-FDG PET Study"

_biomedicines, 2023, doi:10.3390/biomedicines11020506_

Round 1

Reviewer 1 Report

The authors present an interesting study comparing and contrasting the efficacy of 18F-fluoro-deoxy-glucose positron emission tomography (FDG PET) against methods such as MRI in diagnosing autoimmune encephalitis. A retrospective study in nature utilising data of a certain criteria between the years of 2015 and 2020; the authors highlight stark differences in the results/indices measured between the two techniques, while also highlighting the physiological changes that correlated with autoimmune encephalitis in different subpopulations of patients and may act as potential markers of such. Taken together, the authors present strong evidence that FDG PET may represent a viable option for catching cases of autoimmune encephalitis earlier in a clinical setting.

In reviewing the manuscript, I made a few small observations. The following should be addressed when preparing a suitable revision:

1.       The writing of the manuscript is quite strong overall, however there are instances within where the language could be improved/some words switched out in favour of others. The authors should proof-read the manuscript and try to reduce these instances where they occur.

2.       The tables, while containing a lot of information, could be formatted better. The authors need to revise how the data is presented in order to improve how legible the data is and facilitate comparisons between groups.

3.       Out of interest, did the authors perform any kind of analyses with respect to gender on any of the indices measured?

4.       There are a few small typos within that could use attention; for example interchanging how ‘table’ is used (i.e. Table 1 vs. table 2), and using words in place of numbers and vice versa (i.e. forty-three/46 patients presented….)

Reviewer 2 Report

This is a clinical article on a PET study related to glucose metabolism performed at a specialized medical center in France on more than 40 patients with autoimmune encephalitis. The descriptions are clearly written according to style. The population is large and the article is radiologically reliable.

(Major 1) Introduction: As JCM is a general medical journal, it is necessary to start with a definition of encephalitis and a basic knowledge statement about autoimmune encephalitis. A Comprehensive Review of Pediatric Acute Encephalopathy. J Clin Med. 2022 Oct 7;11(19):5921.

(Minor 1) Abstracts: PET is indeed a biomarker, as the authors claim, but it is also imaging. Therefore, how about an imaging biomarker for the diagnosis of autoimmune encephalitis?

(Minor 2) Why did you set your sights on using 18F-FDG for autoimmune encephalitis, as PET with 18F-FDG has already been applied for diagnostic imaging of brain diseases (especially temporal lobe epilepsy and epileptic encephalopathy)? Please mention this point.

(Minor 3) Another known PET for brain diseases is 11C-flumazenil. What is the applicability of PET with other nuclides? Also, there have already been scattered reports of autoimmune encephalitis using 99mTc and 123I-iomazenil SPECT, what do you think?

(Minor 4) There are reports of 18F FDG-PET diagnosing autoimmune encephalitis in children with under 18 as well as in adult cases. Iran J Public Health . 2021 Jan;50(1):203-204. A Case Report of Non-Herpetic Limbic Encephalitis with Psychological Symptoms and Parkinsonism.

(Minor 5) Is it known in autoimmune encephalitis that the clinical differences in subtypes per autoantibody were not statistically significant? Or is this a new finding?

(Minor 6) Have there been any trends in the results of PET studies between men and women?

(Minor 7) Considering surgical therapy, immunosuppressive therapy and steroid therapy, when do the authors consider an MRI or PET scan to be the best test plan?

Best regards,

Dr. Reviewer

Author Response

Pleese see attachment

Round 2

Reviewer 2 Report

Thank you for your prompt reply for reviewer's comments. The authors clearly illustrate the usefulness of FDG-PET testing in differentiating between NMDA-type encephalitis and other types of encephalitis. They also stipulate that the NMDA encephalitis in the paper is for adult cases, and that pediatric NMDA is different. Under these conditions, the paper is assessed as an important reference for medical practitioners. The authors have also adequately responded to the reviewers' comments.